# Genetic and Epigenetic Study of Monozygotic Twins Affected by Parkinson's Disease

Yi-Min Sun [1,†] , Wan-Li Yang [2,†] , Ekaterina Rogaeva [3,4] , Anthony E. Lang [4,5,6] , Jian Wang [1,*] and Ming Zhang [2,7,8,*]

1 Department of Neurology and National Research Center for Aging and Medicine & National Center for Neurological Disorders, Huashan Hospital, Fudan University, Shanghai 200040, China
2 Department of Medical Genetics, The First Rehabilitation Hospital of Shanghai, School of Medicine, Tongji University, Shanghai 200090, China
3 Tanz Centre for Research in Neurodegenerative Diseases, University of Toronto, 60 Leonard Ave., Toronto, ON M5T 2S8, Canada
4 Division of Neurology, University of Toronto, Toronto, ON M5R 0A3, Canada
5 Edmond J. Safra Program in Parkinson's Disease and Morton and Gloria Shulman Movement Disorders Clinic, Toronto Western Hospital, Toronto, ON M5T 2S8, Canada
6 Krembil Brain Institute, Toronto, ON M5G 2C4, Canada
7 Clinical Center for Brain and Spinal Cord Research, Tongji University, Shanghai 200092, China
8 Institute for Advanced Study, Tongji University, Shanghai 200092, China
* Correspondence: wangjian_hs@fudan.edu.cn (J.W.); mingzhang@tongji.edu.cn (M.Z.)
† These authors contributed equally to this work.

**Abstract:** Background: Genetic and epigenetic modifiers of age at onset of Parkinson's disease (PD) are largely unknown. It remains unclear whether DNA methylation (DNAm) age acceleration is linked to age at onset in PD patients of different ethnicities with a similar genetic background. We aim to characterize the clinical, genomic and epigenomic features of three pairs of Chinese monozygotic twins discordant for PD onset by up to 10 years. Methods: We conducted whole genome sequencing, multiplex ligation-dependent probe amplification and genome-wide DNAm array to evaluate the three pairs of Chinese monozygotic twins discordant for age at onset of PD (families A–C). Results: We identified two heterozygous PRKN mutations (exon 2–4 deletion and p.Met1Thr) in PD affected members of one family. Somatic mutation analyses of investigated families did not reveal any variants that could explain the phenotypic discordance in the twin pairs. Of note, our epigenetic study revealed that the twins with earlier-onset had a trend of faster DNAm age acceleration than the later-onset/asymptomatic twins, but without statistical significance. Conclusion: The link between DNAm age acceleration and PD onset in Chinese patients should be interpreted with cautious, and need to be further verified in an extended PD cohort with similar genetic background.

**Keywords:** monozygotic twins; Parkinson's disease; genetics; epigenetics; age at onset

## 1. Introduction

Parkinson's disease (PD) is a clinically heterogenous neurogenerative disease, with up to 80 years difference in age at onset [1]. Genetic modifiers of PD age at onset were largely unclear. DNA methylation and/or environmental exposures [2] were also reported to contribute to variable PD onset. DNA methylation age acceleration (the difference between DNA methylation age and chronological age) measures the cumulative effect on the epigenetic maintenance system and could serve as a biomarker of biological aging. Our recent work suggested that increased DNA methylation age acceleration was significantly associated with earlier PD onset in discovery (*n* = 96) and replication (*n* = 182) idiopathic PD cohorts, as well as 220 *LRRK2* G2019S mutation carriers of Caucasian origin [2]. However, it remains unclear if DNA methylation age acceleration is linked to age at onset in PD

patients of different ethnicities (such as Chinese) with a similar genetic background (e.g., in twin pairs).

Monozygotic twins share almost identical genetic profiles and provide a unique opportunity to clarify genetic and non-genetic modifiers of PD onset. Previous twin studies reported that DNA methylation age acceleration was linked to amyotrophic lateral sclerosis (ALS) discordance in monozygotic twins carrying the same *SOD1* mutation [3] or *C9orf72* repeat expansion [4].

In the current study, we aim to characterize clinical, genomic and epigenomic features in three pairs of Chinese monozygotic twins discordant for PD age at onset, and evaluate whether DNA methylation age acceleration could be underpinning the variation in age at onset in PD patients of Chinese origin.

## 2. Materials and Methods

Three pairs of Chinese monozygotic twins discordant for PD age at onset and their family members were recruited at Huashan Hospital (Shanghai, China). Patients from the families were diagnosed with PD using the UK PD Society Brain Bank Clinical Diagnostic Criteria or MDS Clinical Diagnostic Criteria [5]. Age at onset was defined as the time motor symptoms (tremor, rigidity or bradykinesia) were first reported by the patient. [$^{11}$C]-2β-carbomethoxy-3β-(4-fluorophenyl) tropane (CFT) positron emission computed tomography (PET/CT) was carried out in 2 probands according to a previous report [6].

Collected clinical and demographic information is presented in Table 1. Informed consent was obtained from each study participant in accordance with the ethics review board at Huashan hospital. The present study conforms to the tenets of the Declaration of Helsinki and was approved by the Ethics Committee of the Huashan Hospital, Fudan University (Shanghai, China).

The monozygotic status of the three pairs of twins were confirmed by genotyping polymorphic markers (Supplementary Table S1) using the STRtyper-21G kit (Health Gene Technologies Co., Ltd.; Ningbo, China).

We used the PCR-free library and Novaseq 6000 analyzer (Illumina, San Diego, CA, USA) for whole genome sequencing study. The sequencing reads were mapped against the human genome reference assembly (UCSC Genome Browser hg19) using Burrows-Wheeler-Alignment software. We used Samtools and Picard tools to generate the BAM files; and the Genome Analysis Toolkit (GATK4.2.2.0) pipeline (https://gatk.broadinstitute.org/hc/en-us, accessed on 15 August 2022) to call single nucleotide variants (SNVs) or insertion/deletions (InDels). Filtered exonic and splicing variants were further annotated with ANNOVAR. To select variants of interest, we filtered SNVs/InDels with a minor allele frequency (MAF) < 0.001 in the Genome Aggregation Database (v2.1.1) of East Asian origin (gnomAD_EAS; $n$ = 780) and China Metabolic Analytics Project database (ChinaMAP; $n$ = 10,588). We also used in silico tools (Sorting Tolerant From Intolerant (SIFT), PolyPhen-2 and MutationTaster) to predict the functional consequences of candidate variants. We also selected SNVs with a Combined Annotation Dependent Depletion score above 10 representing the top 10% of deleterious variants in the human genome. We investigated whether deleterious variants map to any reported PD risk genes (Supplementary Table S2). When the pathogenicity of the genetic variant was undetermined, it was rated according to the 2015 American College of Medical Genetics and Genomics guidelines [7]. *PRKN* variants were confirmed by Sanger sequencing (Supplementary Table S3).

To interrogate PD-related copy number variations in *PARK7*, *PRKN*, *SNCA*, *PINK1*, *LRRK2*, *ATP13A2*, *GCH1* and *UCHL1*, we also performed MLPA by a SALSA® MLPA probe mix P051-D1/P052-D2 Parkinson kit (MRC-Holland, the Netherlands). ExpansionHunter (version 4.0.2) was used to analyze short tandem repeats (STRs) in 28 genes causing neurological diseases (Supplementary Table S4).

**Table 1.** Clinical features of three families with identical twins discordant for age at onset of Parkinson's Disease.

| Clinical Features | Family A | | Family B | | | Family C | |
|---|---|---|---|---|---|---|---|
| | II-2 | II-3 | II-3 | II-5 | II-6 | II-1 | II-2 |
| Sex | Female | Female | Female | Female | Female | Female | Female |
| PD AAO, years | - | 54 | 36 | 40 | 50 | 46 | - |
| ASC, years | 55 | 55 | 54 | 51 | 51 | 52 | 52 |
| AOA, years | 55 | 55 | 54 | 51 | 51 | 52 | 52 |
| Types of diet | Normal | Normal | Normal | Normal | Normal | Normal | Normal |
| Occupation | Doctor | Retired cashier of a pesticide and chemical fertilizer company | NA | Worker | Worker in a shoe factory | Civil servant | Civil servant |
| Chemical toxins exposure | No | pesticide and chemical fertilizer | NA | No | No | carbon monoxide poisoning at age 45 | No |
| Head trauma | No | No | NA | No | No | Yes | No |
| Surgery with general anesthesia | No | No | NA | No | No | Ovariectomy | No |
| Cigarette smoking | No | No | NA | No | No | No | No |
| Alcoholic consumption | No | No | NA | light | light | No | No |
| Physical activity in leisure time | No | No | No | No | No | No | No |
| Exposure to pathogens or infectious agents | No | No | No | No | No | No | No |
| Other medical history | No | No | NA | lumbar disc herniation | lumbar disc herniation | No | No |
| Cardinal symptoms of PD | - | Bradykinesia, rigidity | Tremor, Bradykinesia, rigidity | Tremor, Bradykinesia, rigidity | Tremor, Bradykinesia, rigidity | Tremor, Bradykinesia, rigidity | - |
| Family history of PD | No | No | AR | AR | AR | AD | AD |
| Years of education | NA | 10 | 6 | 3 | 3 | 12 | - |
| LEDD, mg | - | NA | 451 | 150 | 250 | 800 | - |
| H&Y stage | - | 2 | 2 | 2 | 2 | 3 | - |
| MMSE score | - | NA | NA | 24 | 28 | NA | - |
| UPDRS-III score | - | 30 (Med-off) | 29 (Med-on) | NA | 33 (Med-off) | 35 (Med-on) | - |
| Methods of genetic analysis | MLPA + WGS | MLPA + WGS | MLPA | MLPA + WGS | MLPA + WGS | MLPA + WGS | WGS |
| Results of genetic analysis | exon 1–3 del (het) in *PRKN* | exon 1–3 del (het) in *PRKN* | Exon 2–4 del (het), c.2T > C, p.Met1Thr (het) (NM_004562.3) in *PRKN* | Exon 2–4 del (het), c.2T > C, p.Met1Thr (het) (NM_004562.3) in *PRKN* | Exon 2–4 del (het), c.2T > C, p.Met1Thr (het) (NM_004562.3) in *PRKN* | - | - |
| [11]C-CFT PET/CT results | NA | Decreased DAT uptake ratios in bilateral caudate and putamen, especially in the left side | NA | NA | NA | Decreased DAT uptake ratios in bilateral caudate and putamen, especially in the left side | NA |

AAO: age at onset; AD: autosomal dominant inheritance mode; AOA: age of assessment; AR: autosomal recessive inheritance mode; ASC: age at sample collection; [11]C-CFT PET/CT: [11C]-2β-carbomethoxy-3β-(4-fluorophenyl)tropane positron emission computed tomography; H&Y: Hoehn & Yahr scale; LEDD: Levodopa equivalent daily dosage; Med-off: off medication; Med-on: on medication; MMSE: Mini-mental state examination; MLPA: multiplex ligation-dependent probe amplification; NA: not available; PD: Parkinson's disease; UPDRS-III score: Unified Parkinson's disease rating scale (UPDRS) motor examination (items 18–31); WGS: whole genome sequencing.

To explore if somatic mutations might explain the PD discordance, we used Mutect2 to call somatic variants by comparing the BAM files of the monozygotic twins in each twin-pair. Putative somatic mutations were selected according to a reported strategy [8]. In brief, we set the filtering parameters to exclude variants as follows: (1) an allele depth at candidate site < 10; (2) variants with a low consensus confidence score (variant confidence divided by an unfiltered depth < 2); (3) mutations with a MAF of >0.01 in gnomAD database; (4) variants with co-occurrence in both twins. Candidate variants were further annotated by ANNOVAR. The filtered coding or splicing variants were further analyzed by Sanger sequencing.

We profiled the DNA methylation levels of >85,000 CpGs in blood genomic DNA samples using the Infinium Methylation EPIC BeadChip (Illumina) at Novogene Co., LTD (Beijing, China). Raw DNA methylation data derived from EPIC array (Illumina) was extracted, and quantile preprocessed by Illumina GenomeStudio (Version 2011.1) and the minfi package in R (version 4.0.4). The beta (β) value was used to estimate the DNA methylation level of each CpG by the intensity ratio between the methylated and the sum of

methylated and unmethylated sites. DNA methylation age was estimated using the DNA methylation age calculator (https://dnamage.genetics.ucla.edu, accessed on 20 September 2022). DNA methylation age acceleration is the difference between DNA methylation age and chronological age. The association between the DNA methylation status of a single CpG and discordant PD onset was analyzed using the linear regression model of the R minfi package.

We used the Wilcoxon rank sum test (R 4.0.4) to analyze the association between DNA methylation differences and PD status. *p*-value < 0.05 was accepted as statistically significant.

## 3. Results

Clinical information of the investigated PD families is presented in Table 1 and Figure 1. The three families were discordant for PD onset by 1, 6 and 10 years. We observed a history of neurotoxin exposure in two affected twins. In family A, the proband (II-3) had PD from age 54 with a history of pesticide and chemical fertilizer exposure. Her asymptomatic twin sister (II-2) has not been exposed to pesticides or chemical fertilizers. In family C, the proband (II-1) developed PD at age 46 and had exposure to carbon monoxide (CO) poisoning, head trauma and general anesthesia. The asymptomatic twin (II-2) presented with no motor symptoms 6 years after the proband (II-1) developed PD. Their mother also had PD with an age at onset > 60. In family B, the twin-pair had a PD age at onset discordance of 10 years (age 40 vs. 50) with a similar environmental exposure. The family also includes an affected sibling (II-3) with an age at onset of 36. Both the twins lived nearby in the same city or town.

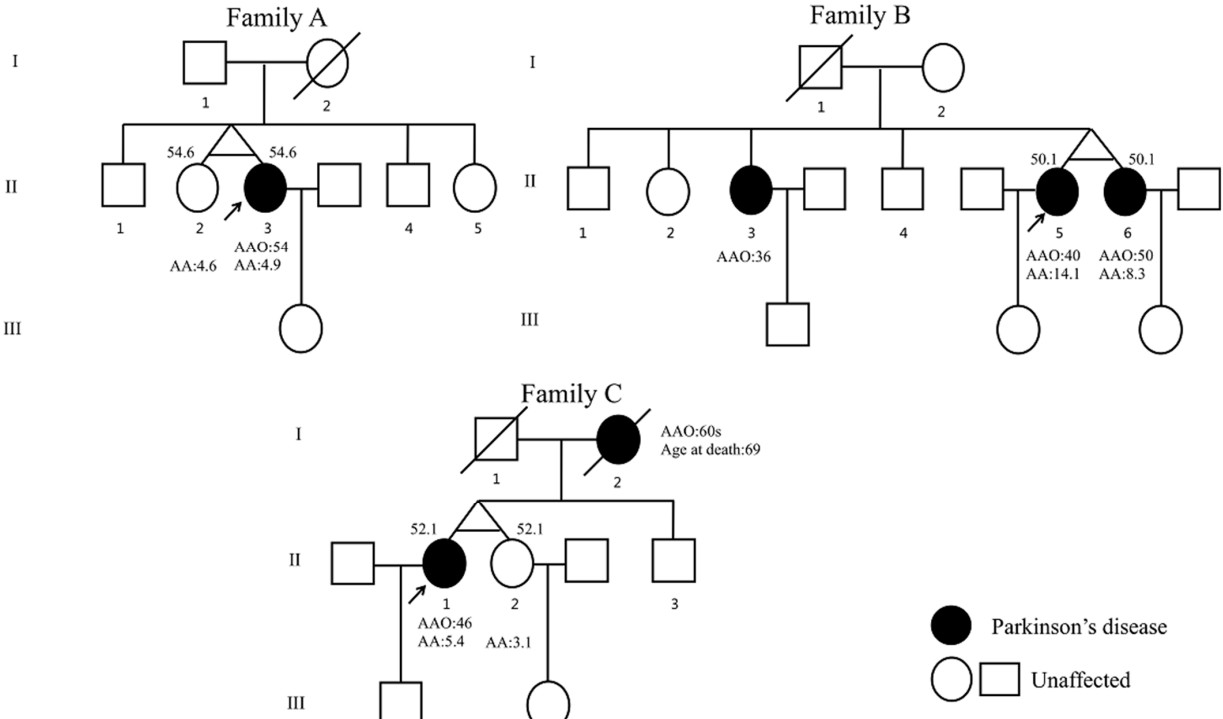

**Figure 1.** Pedigrees of three monozygotic twin pairs affected by PD. Age at time of sample collection is shown above the symbol. AAO: age at onset; AA: DNA methylation age acceleration.

The cerebral magnetic resonance images of the patients (II-3 in family A, II-5, II-6 in family B, and II-1 in family C) were normal. The $^{11}$C-CFT PET/CT results of the proband in families A and C showed decreased dopamine transporter (DAT) uptake ratios in bilateral caudate and putamen, more decreased on the left side, which correlated with the more prominent symptoms of parkinsonism on the right side and supported the diagnosis of PD (Figure 2).

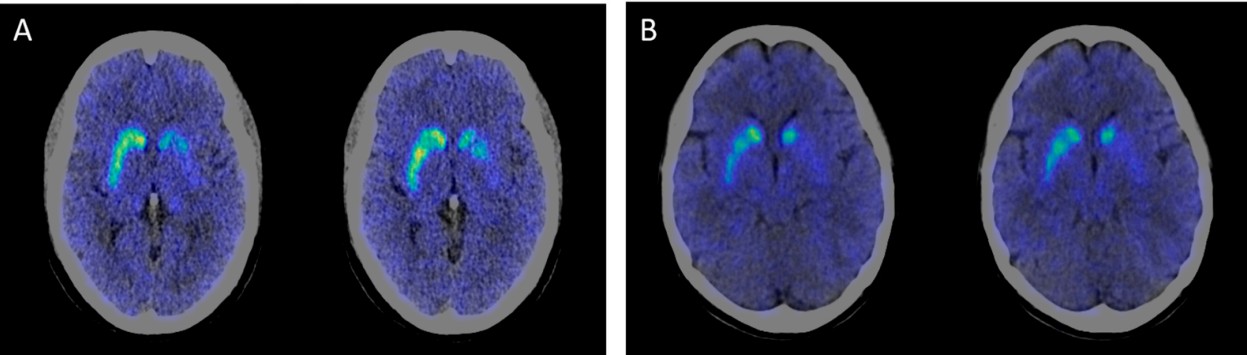

**Figure 2.** The $^{11}$C-CFT PET/CT results. (**A**) The $^{11}$C-CFT PET/CT of proband II-3 in family A showed decreased DAT uptake ratios in bilateral putamen, especially on the left side, and slightly decreased DAT uptake ratios in the left caudate; (**B**) The $^{11}$C-CFT PET/CT of proband II-1 in family C was similar to II-3 of family A.

To understand the genetics of the PD twin pairs, we investigated germline genetic variants mapped to 30 known PD genes (Supplementary Table S2) and 113 GWAS PD risk loci [9] in the families. We identified a heterozygous deletion of exon 2–4 and a heterozygous start loss mutation (NM_004562.3, c.2T > C, p.Met1Thr, rs771586218) in *PRKN* (Table 1, Supplementary Figure S1) in PD patients of family B (II-5, II-6 and II-3). Both variants were known pathogenic. The compound heterozygous status of these two variants could not be verified, since the parents and other siblings declined genetic testing.

Whole genome sequencing revealed no known pathogenic SNVs/InDels or neurological disease-causing STRs (Supplementary Table S4) segregating with disease presentation in two families (Family A or Family C). In family A, we identified a heterozygous *PRKN* exon 1–3 deletion in the monozygotic twins (II-3 and II-2), but their unaffected father (I-1, 83-years old) carried the same variant (Supplementary Figure S1). In family C (II-1 and II-2), we did not identify any candidate genetic variants in the PD-related genes.

To assess if somatic mutations may explain the disease discordant in the twin pairs, we conducted a somatic mutation analysis and we found no differential somatic SNVs or InDels mapping to 30 PD genes or 113 PD GWAS risk loci (Supplementary Table S5), suggesting that known PD-risk genetic variants may not explain the discordant onset. Of note, we observed five rare variants in family A and one rare variant in family B (Supplementary Table S6), however none of them were validated by Sanger sequencing (Supplementary Figure S2).

To test if DNA methylation age acceleration is linked to discordant PD onset in Chinese monozygotic twins, we performed a genome-wide DNA methylation analysis. We found that the affected/earlier onset twin had a trend of higher DNA methylation age acceleration (4.9 vs. 4.6 years in family A, 14.1 vs. 8.3 years in family B and 5.4 vs. 3.1 years in family C (Figure 1, Supplementary Figure S3) but without statistical significance. In addition, a single CpG analysis did not reveled any significant differences in the DNA methylation level between the twins discordant for the PD age at onset. Only one CpG site (cg16681355) showed a trend of a difference in DNAm levels ($p = 0.1$, delta beta > 15%).

## 4. Discussion

We performed a clinical, genetic and epigenetic study of three Chinese monozygotic twin pairs discordant for PD onset. The genetic features were identical between each pair of twins, while the environmental exposures and a trend (with no statistical significance) of faster DNA methylation age acceleration in the earlier onset/affected monozygotic twins was found, indicating the environmental exposures and DNA methylation related biological aging might modify PD onset.

Genomes of MZ twins may not be 100% identical, because genome/exome studies have discovered several MZ twins who have discordant mutations, which arose in the

soma after fertilization. For instance, some studies identified somatic mutations that might explain psychiatric disease discordance in MZ twins [10,11]. However, our results did not suggest that somatic variants are linked to the phenotypic discordance of investigated identical twins. Of note, somatic mutations were also not found in identical twins discordant for another neurodegenerative disease (ALS) [12].

The search for germline mutations in known PD-genes did not reveal any mutations segregating with PD in family A and C. Of note, in family A we identified a heterozygous *PRKN* exon 1–3 deletion in the MZ twins and their unaffected father. Single heterozygous deletions are not considered to be sufficient to cause PD [13]. *PRKN* is linked to autosomal recessive PD with compound heterozygous and homozygous mutations (e.g., point mutations and exon rearrangements) have been identified in up to 50% familial PD patients [14].

We also identified two known pathogenic heterozygous *PRKN* variants in family B. It would suggest that epigenetic factors might modify disease onset in monogenetic forms of PD if they could be confirmed the compound heterozygous status, which was unfortunately not in this study.

The affected patients in Family A had the environmental exposure of pesticides, which is well known neurotoxins that may confer risk of PD onset [15]. Pesticides exposure was found increasing PD risk by 1.7 folds in a large cohort with longitudinal follow-up [16]. Different pesticides might contribute to PD by different mechanism, but the kind of pesticide the patient in Family A exposed to, the exposure frequency, and the exposure time could not be reported by the patient. We inferred the pesticides exposure might attribute to PD onset.

We noticed the affected twin in family C developed PD one year after the CO intoxication and the CO related parkinsonism should be excluded [17]. The patient visited our center 4 years before the blood collecting with UPDRS (Med-off) score of 16. The disease progression [18] and the decreased DAT uptake ratios in putamen different from the pattern in CO related CO related parkinsonism (predominant in caudate regions) indicated the PD diagnosis. Since CO intoxication patients had 9-fold increased risk of PD [19] and the hazard ratio of PD was significantly higher in 2 years after CO poisoning than after the first 2 years, CO poisoning might also contribute to her PD onset. Additionally, both pesticide exposure and CO poisoning may affect biological aging status (DNA methylation age) through epigenetic changes of several genes [20,21]. But whether CO poisoning affect the DNA methylation age in the patient was not known since the time it took to have DNA methylation age affected was unknown.

The unaffected twin in family A and C reported no motor symptoms of PD, but we did not perform the UPDRS-III or a DAT $^{11}$C-CFT PET/CT scan at the time of blood extraction. The non-motor features could be assessed in the future to evaluate whether they were in the prodromal stage of PD [22,23].

Our epigenetic study revealed that the twin with the earlier-onset had a faster DNAm age acceleration than the twin with the later-onset (or asymptomatic twin), in which the twin with the earlier-onset had increased DNAm age acceleration. DNAm age acceleration could serve as a biomarker of biological aging and can measure the cumulative effect on the epigenetic maintenance system [24]. DNAm age acceleration has been associated with disease presentation in several neurodegenerative diseases [25]. For example, increased DNAm age acceleration was reported to be linked with earlier AAO in sporadic ALS [26], *C9orf72* related ALS/FTD [27], and *SNCA* p.A53E related PD [28], as well as with the risk of sporadic PD [29]. DNAm age acceleration was also reported to be associated with survival and disease duration in ALS patients [26,27]. Previously, DNAm age acceleration was associated with discordant ALS onset in MZ twins carrying the same *SOD1* mutation [3], or *C9orf72* repeat expansion [4]. The current study indicates the notion that DNAm-related biological aging may modify disease onset of neurodegenerative diseases.

Our recent work in the Parkinson's Progression Markers Initiative (PPMI) and Canadian PD cohorts identified a link between DNA methylation age acceleration and age at

onset in sporadic patients and *LRRK2* carriers [2]. Our current findings were in line with the results but with no statistical significance which might be due to the small sample size. Future studies could include more monozygotic twin pairs to clarify the roles of environmental/epigenetic factors in modifying PD onset. In addition, autopsy tissues of discordant twins could be used in the future to investigate if somatic mutations in brain tissues could explain phenotype discordance.

The current study is limited to a small sample set of three monozygotic twin pairs of Chinese origin. Future studies should include more monozygotic twin pairs or a larger PD cohort with similar genetic background (e.g., *PRKN* mutation carriers) to clarify the role of environmental/epigenetic factors in modifying PD onset, and the link between DNA methylation age acceleration and disease subtypes or progression.

Our study in Chinese twin pairs showed a trend of increased DNA methylation age acceleration in the PD affected/earlier onset twins but without statistical significance. We should be cautious to interpret the link between DNA methylation age acceleration and PD onset in Chinese patients because of the limited sample size. Further studies should include more Chinese patients with similar genetic background to clarify the role of DNA methylation age acceleration.

**Supplementary Materials:** The following supporting information can be downloaded at: https://www.mdpi.com/article/10.3390/ctn7020011/s1, Figure S1: Genetic findings for the MZ twins and their family members; Figure S2: Sanger sequencing did not validate somatic mutations in family A and B; Figure S3: Scatter plot of the DNAm age acceleration; Table S1: Haplotype results for the three twin pairs; Table S2: List of known PD risk genes; Table S3: List of primers used to confirm the *PRKN* variants; Table S4: Repeat length of 28 pathogenic STRs in three identical twin pairs; Table S5: SNVs of 113 Parkinson's disease related genes in three MZ twin pairs; Table S6: Mutect2 analysis suggested rare somatic SNVs in family A and B.

**Author Contributions:** Conceptualization, M.Z. and J.W.; methodology, W.-L.Y. and Y.-M.S.; validation, Y.-M.S., W.-L.Y. and M.Z.; data curation, Y.-M.S., J.W., A.E.L. and E.R.; writing—original draft preparation, Y.-M.S., W.-L.Y. and M.Z.; writing—review and editing: J.W., M.Z., A.E.L. and E.R. All authors have read and agreed to the published version of the manuscript.

**Funding:** This research was partially funded by the National Natural Science Foundation of China (grant number 82071430) (M.Z.), (grant number 82171421, 91949118) (J.W.), National Health Commission of China (grant number Pro20211231084249000238) (J.W.), Shanghai Municipal Natural Science Foundation General Program (grant number 22ZR1466400) (M.Z.), Shanghai Municipal Science and Technology Major Project (grant number 2018SHZDZX01 and 21S31902200) (J.W.), ZJ Lab and Shanghai Center for Brain Science and Brain-Inspired Technology (J.W.), the Fundamental Research Funds for the Central Universities (M.Z.), the Canadian Consortium on Neurodegeneration in Aging (E.R.) The other authors don't have any financial disclosures.

**Institutional Review Board Statement:** The study was conducted in accordance with the Declaration of Helsinki, and approved by the Ethics Committee at Huashan Hospital (protocol code 2011-174-4, 20 February 2021).

**Informed Consent Statement:** Informed consent was obtained from all subjects involved in the study. Patients have signed consent for their data to be used for the purposes of research. None of the data reported in this manuscript can be used to identify the patients presented.

**Data Availability Statement:** The data presented in this study are available upon reasonable request from the corresponding authors.

**Acknowledgments:** We would like to thank the patients and their family members who participated in this study. None of study participants received financial incentives. This work was partially supported by the National Natural Science Foundation of China (82071430) (M.Z.), (82171421, 91949118) (J.W.), National Health Commission of China (Pro20211231084249000238) (J.W.), Shanghai Municipal Natural Science Foundation General Program (22ZR1466400) (M.Z.), Shanghai Municipal Science and Technology Major Project (2018SHZDZX01 and 21S31902200) (J.W.), ZJ Lab and Shanghai Center for Brain Science and Brain-Inspired Technology (J.W.), the Fundamental Research Funds for the Central Universities (M.Z.), the Canadian Consortium on Neurodegeneration in Aging (E.R.).

**Conflicts of Interest:** The authors declare no conflict of interest.

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
