# Peer review of "Genetic and Epigenetic Study of Monozygotic Twins Affected by Parkinson’s Disease"

_ctn, doi:10.3390/ctn7020011_

Round 1
Reviewer 1 Report
Very interesting observation. However, sample size is too less to conclude the link between DNAm age and onset of PD. There might be ethnic differences, but conclusion can not drawn based on limited sample size.
Introduction require much better and concise representation. There are too many odd abbreviation that may be reduced like DNAm, AAO, MZ etc. Last para of the discussion require appropriate citation.
Author Response
Response to Reviewer 1 Comments
Point 1: Very interesting observation. However, sample size is too less to conclude the link between DNAm age and onset of PD. There might be ethnic differences, but conclusion can not drawn based on limited sample size.
Response 1: We appreciated the reviewer for the concern. We rewrote the conclusion in the revised version (Last paragraph in discussion).
Point 2: Introduction require much better and concise representation. There are too many odd abbreviation that may be reduced like DNAm, AAO, MZ etc. Last para of the discussion require appropriate citation.
Response 2: According to the reviewer’s concern, we replace “MZ” with “monozygotic”, “AAO” with “age at onset”, “DNAm” with “DNA methylation“ in the manuscript. We rewrote the last paragraph in discussion as a conclusion (Last paragraph in discussion).
Please see the attachment

Reviewer 2 Report
The present study investigates genetic and epigenetic modifiers of age at onset (AAO) of Parkinson’s disease on three pairs of Chinese monozygotic (MZ) twins. The study provides substantial clinical, genetic and epigenetic data that are relevant to the understanding of discordance of AAO in PD and the role of environmental factors for PD onset. However, due to the small size of the patient cohort, the link between epigenetic and AAO in three pairs MZ twins affected with PD has not formally established. Further studies with larger cohorts seem to be warranted to clarify the role of environmental factors, epigenetic (DNA methylation) and AAO in PD.
I have a few questions and minor points for the discussion:
1) The authors identified in family A that the proband II-3 had a history of pesticide and chemical fertilizer exposure. This is an important observation suggesting the implication of environmental exposure to acceleration disease onset. However, the authors could not correlate that with DNA methylation levels. Are the chemical agents that the proband II-3 were exposed to known to increase PD risk (such as rotenone, paraquat, TCE…)? How long were the periods of exposure? How frequent were the exposures?
2) For all three families: The reviewer assumes that the patients have been selected based on the criteria of similar environmental exposure. Please provide some additional information, at the best of the author knowledge, regarding the patient's history of exposure to pathogens or infectious agents, types of diet, levels of exercise...? Are there any differences in their location of residence that may affect their exposome?
3) The authors identified heterozygous deletion of exon 2-4 PRKN in family B. Are these mutations known as PD risk/pathogenic mutations?
4) The authors discuss that CO poisoning may confer risk of PD onset for the family C. However, the reviewer notes that the poisoning of the patient II-1 happened at age of 45, only one year prior the PD onset. What is known about frequency or levels of CO poisoning and PD risk? May the authors clarify it further in the discussion.
5) The authors seem make two contradictory statements in the discussion: “…which revealed a trend but not significantly faster DNAm age acceleration in the earlier onset/affected MZ twins. This is in line with our recent finding in the Parkinson's Progression Markers Initiative (PPMI) and Canadian PD 186 cohorts that identified a link between DNAm age acceleration and AAO in sporadic patients and LRRK2 carriers” vs “Our study in Chinese twin pairs showed a trend of increased DNAm age acceleration in the PD affected/earlier onset twins but without statistical significance, which is not fully in line with our recent findings in idiopathic PD patients and LRRK2 carriers of Caucasian origin”. Please clarify!
Author Response
The present study investigates genetic and epigenetic modifiers of age at onset (AAO) of Parkinson’s disease on three pairs of Chinese monozygotic (MZ) twins. The study provides substantial clinical, genetic and epigenetic data that are relevant to the understanding of discordance of AAO in PD and the role of environmental factors for PD onset. However, due to the small size of the patient cohort, the link between epigenetic and AAO in three pairs MZ twins affected with PD has not formally established. Further studies with larger cohorts seem to be warranted to clarify the role of environmental factors, epigenetic (DNA methylation) and AAO in PD.
I have a few questions and minor points for the discussion:
Point 1: The authors identified in family A that the proband II-3 had a history of pesticide and chemical fertilizer exposure. This is an important observation suggesting the implication of environmental exposure to acceleration disease onset. However, the authors could not correlate that with DNA methylation levels. Are the chemical agents that the proband II-3 were exposed to known to increase PD risk (such as rotenone, paraquat, TCE…)? How long were the periods of exposure? How frequent were the exposures?
Response 1: We appreciated the reviewer for the concern. We did ask the patient the kind of pesticide, the frequency and the time of exposure, but she could not report it clearly. We could only infer the PD onset might be related to the pesticide exposure. We discussed it in the discussion (4th paragraph in discussion)
Point 2: For all three families: The reviewer assumes that the patients have been selected based on the criteria of similar environmental exposure. Please provide some additional information, at the best of the author knowledge, regarding the patient's history of exposure to pathogens or infectious agents, types of diet, levels of exercise...? Are there any differences in their location of residence that may affect their exposome?
Response 2: We appreciated the reviewer for the suggestion. We telephoned the patient or the family members and found they do not have the history of exposure to pathogens or infectious agents. The types of diet was normal Chinese food. They both lived nearby in the same city or town. They did not do extra exercises. We added the information in the Table (Table 1) and Result (last line in 1st paragraph in result)
Point 3: The authors identified heterozygous deletion of exon 2-4 PRKN in family B. Are these mutations known as PD risk/pathogenic mutations?
Response 3: We appreciated the reviewer for pointing out this. Both variants (p.Met1Thr and exon 2-4 deletion) were known pathogenic variants. We made it clear in the revised version (Line 5, 3rd paragraph in Result).
Point 4:The authors discuss that CO poisoning may confer risk of PD onset for the family C. However, the reviewer notes that the poisoning of the patient II-1 happened at age of 45, only one year prior the PD onset. What is known about frequency or levels of CO poisoning and PD risk? May the authors clarify it further in the discussion.
Response 4: We appreciated the reviewer for the concern. She had CO intoxication in age 45, one year before the PD onset. The CO intoxication might increase the PD risk and contribute to the PD onset. We explained in the discussion (5th paragraph in discussion).
Point 5: The authors seem make two contradictory statements in the discussion: “…which revealed a trend but not significantly faster DNAm age acceleration in the earlier onset/affected MZ twins. This is in line with our recent finding in the Parkinson's Progression Markers Initiative (PPMI) and Canadian PD 186 cohorts that identified a link between DNAm age acceleration and AAO in sporadic patients and LRRK2 carriers” vs “Our study in Chinese twin pairs showed a trend of increased DNAm age acceleration in the PD affected/earlier onset twins but without statistical significance, which is not fully in line with our recent findings in idiopathic PD patients and LRRK2 carriers of Caucasian origin”. Please clarify!
Response 5: We appreciated the reviewers for pointing this out for us. We are sorry that we did not make it clear. We found increased DNA methylation age acceleration significantly associated with an earlier PD onset in 96 Caucasian idiopathic PD patients and 220 LRRK2 G2019S mutation carriers of Caucasian population (PMID: 35921480). Though we revealed a trend of faster DNA methylation age acceleration in earlier onset/affected monozygotic twins, no statistical significance was reached. The limited sample size is the shortcoming of the study. We rewrote the paragraphs containing the two statements in the discussion (1st paragraph, and last paragraph in discussion).
Please see the attachment
